# Anti-correlation of LacI association and dissociation rates observed in living cells

Vinodh Kandavalli ⬛, Spartak Zikrin ⬛, Johan Elf ⬛ ✉ & Daniel Jones ⬛ ✉

The rate at which transcription factors (TFs) bind their cognate sites has long been assumed to be limited by diffusion, and thus independent of binding site sequence. Here, we systematically test this assumption using cell-to-cell variability in gene expression as a window into the in vivo association and dissociation kinetics of the model transcription factor LacI. Using a stochastic model of the relationship between gene expression variability and binding kinetics, we performed single-cell gene expression measurements to infer association and dissociation rates for a set of 35 different LacI binding sites. We found that both association and dissociation rates differed significantly between binding sites, and moreover observed a clear anticorrelation between these rates across varying binding site strengths. These results contradict the long-standing hypothesis that TF binding site strength is primarily dictated by the dissociation rate, but may confer the evolutionary advantage that TFs do not get stuck in near-operator sequences while searching.

Cell-to-cell variability in gene expression, sometimes called "noise," is a fact of life in the low molecular copy number regime in bacterial cells. A variety of mechanisms underpin this variability, including the stochastic binding and unbinding of transcription factors (TFs)[1] and RNA polymerase (RNAP)[2], gene dosage effects, and partitioning of macromolecules upon cell division[3,4], among many others[5,6]. Conversely, measurements of gene expression variability can be used to shed light on these cellular processes[7–9], although attributing observed variability to specific mechanisms can be challenging. Among the biological implications of gene expression variability, one of the most prominent is "bet-hedging," the idea that subpopulations of cells exhibiting altered gene expression states may be better prepared to survive rapid and unpredictable shifts in environmental conditions[10,11].

In this work, we focus on variability due to the stochastic association and dissociation of TFs. A long-standing assumption has been that TF association is diffusion-limited: that is, that the association rate is largely independent of the strength of the TF binding site[7,12]. Under this assumption, the strength (reflected in the dissociation constant $K_d$) of a particular binding site is modulated by changes in the dissociation rate $k_d$. However, recent in vitro experiments have called this assumption into question, showing that the dissociation rate $k_d$ and association rate $k_a$ were negatively correlated for the model TF LacI[13].

Here, we use cell-to-cell variability in gene expression as a window into LacI kinetics in living cells. By modeling the relationship between variability and LacI kinetics, we use single-cell gene expression measurements to infer association and dissociation rates for a set of 35 different LacI binding sites. Consistent with recent in vitro results, we find a clear anticorrelation between association and dissociation rates, upending long-standing understandings of in vivo TF binding kinetics.

## Results

We designed a set of promoter-operator constructs in which LacI binding sites were placed immediately downstream of the apFAB120 promoter (Supplementary Fig. 1), such that transcription from the promoter is occluded when LacI is bound, and transcription is enabled when LacI is not bound (Fig. 1a). To relate the observed mRNA copy number distributions to LacI association and dissociation rates, we used the standard two-state model of stochastic gene expression, sometimes referred to as the "random telegraph" model[14,15] (Fig. 1a). In this model, the promoter can be found in one of two states depending on whether LacI is bound to its operator. In the "on" state, corresponding to unbound LacI, transcripts are produced at rate $r$; in the "off" state, corresponding to bound LacI, transcription is blocked. In both states, mRNAs are degraded at rate $\gamma$. Transitions from "off" to

Science for Life Laboratory, Department of Cell and Molecular Biology, Uppsala University, Uppsala, Sweden. ✉e-mail: johan.elf@icm.uu.se; daniel.jones@icm.uu.se

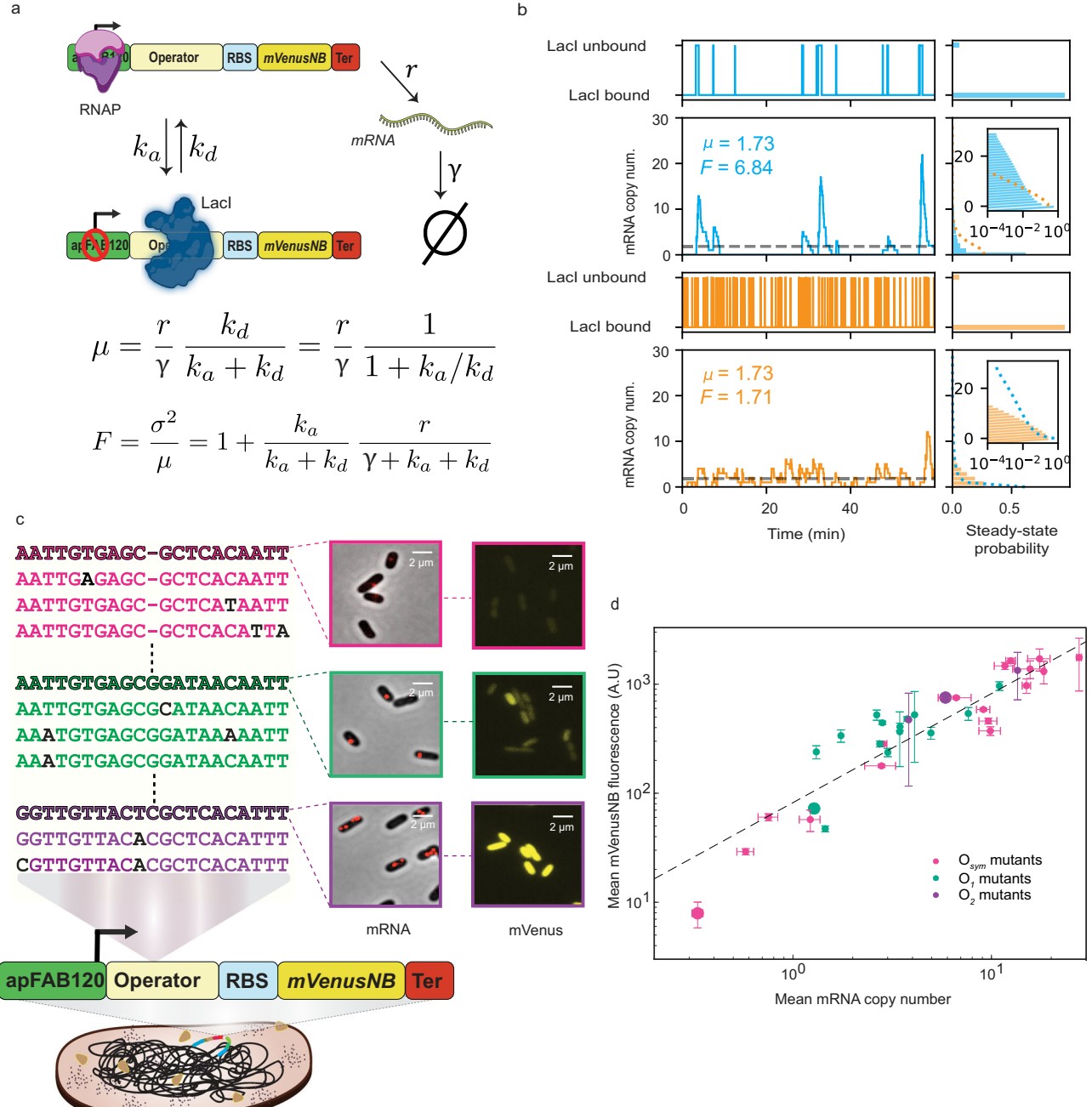

**Fig. 1 | Promoter-operator construct and model relating gene expression variability to transcription factor kinetics. a** The construct consists of a LacI binding site ("Operator") immediately downstream of the apFAB120 promoter, driving expression of the mVenusNB fluorescent protein. The system is either in a LacI-unbound state (top) in which mRNA is produced at rate $r$, or a LacI-bound state in which transcription is halted (bottom), (created in BioRender https://BioRender.com/q45e576). mRNA is degraded at rate $\gamma$, and $k_a$ and $k_d$ represent the LacI association and dissociation rates, respectively. Expressions for mean expression $\mu$ and Fano factor $F$ as a function of model parameters are shown below the schematic[14]. **b** Stochastic simulations of mRNA production for "slow" (top, blue; $k_a = 3.0$ min$^{-1}$, $k_d = 0.18$ min$^{-1}$) and "fast" (bottom, orange; $k_a = 30$ min$^{-1}$, $k_d = 1.8$ min$^{-1}$) operators. $r = 25$ min$^{-1}$ and $\gamma = 0.8$ min$^{-1}$ for both operators. Both operators have the same steady-state LacI binding probability and hence the same mean mRNA expression. However, the "slow" operator exhibits greater variability,

reflected in its larger Fano factor and longer-tailed steady-state mRNA copy number distribution. For easier comparison, the "fast" mRNA distribution is plotted (orange dashed line) on top of the "slow" mRNA distribution (blue bars) and vice versa. The mRNA distributions are plotted on a log scale in insets. **c** Subset of operator mutants assayed in this paper. Operators used the $O_{sym}$ (pink), $O_1$ (green), or $O_2$ (purple) operators as starting points. The wild-type operators are shown in bold and mutations are highlighted in black. Gene expression levels as measured by mRNA FISH ("mRNA") and fluorescence microscopy ("mVenusNB") are shown for $O_{sym}$, $O_1$, and $O_2$. **d** For each operator, the mean mRNA copy number is plotted against the mean mVenusNB fluorescence, showing (as expected) a strong correlation. $O_{sym}$, $O_1$, and $O_2$ wild-type operators are plotted with larger circles; error bars represent sem from bootstrapping. Each data point corresponds to at least 862 individual cells assayed across at least two biological replicates. Source data are provided as a Source Data file.

"on" occur at rate $k_d$, corresponding to LacI dissociation, whereas transitions from "on" to "off" occur at rate $k_a$, corresponding to LacI association. Note: in this work, $k_a$ should be interpreted as an observed empirical rate constant under conditions of constant LacI concentration; hence $k_a$ has units of min$^{-1}$ rather than min$^{-1}$ M$^{-1}$ or s$^{-1}$ M$^{-1}$ as with a true association rate constant.

A full analytic solution to the resulting chemical master equation has previously been derived (see section "Methods")[14,15]. Under this model, the mean mRNA expression is given by

$$\mu = \frac{r}{\gamma} \frac{k_d}{k_a + k_d} = \frac{r}{\gamma} \frac{1}{1 + k_a/k_d}, \quad (1)$$

where $\frac{r}{\gamma}$ is the mean mRNA copy number in the absence of LacI while the second term is simply the fraction of time that LacI is not bound. Notably, the mean expression is dependent on the ratio of association rate to dissociation rate, but does not depend on their absolute magnitude.

To obtain the absolute magnitudes of $k_a$ and $k_d$, we need to consider higher moments of the distribution. It is convenient to use the Fano factor, defined as the variance $\sigma^2$ divided by the mean $\mu$. For this model, the Fano factor is given by

$$F = \frac{\sigma^2}{\mu} = 1 + \frac{k_a}{k_a + k_d} \frac{r}{\gamma + k_a + k_d}. \quad (2)$$

In Fig. 1b, stochastic simulations of two hypothetical operators are shown: one operator with slow association and dissociation kinetics (top, blue), and one operator with fast kinetics (bottom, orange), as well as the corresponding steady-state distributions (Fig. 1b, right). Both operators have the same $k_a$ to $k_d$ ratio, resulting in the same fractions of time spent in the "on" (LacI unbound) and "off" (LacI bound) states and the same mean expression. However, the slow operator is characterized by infrequent bursts of large numbers of transcripts, whereas the fast operator maintains a more constant production. This difference is reflected in a broader steady-state distribution and a larger Fano factor value for the slow operator.

To systematically investigate the relationship between association and dissociation rates for LacI binding sites, we selected a set of 35 binding sites to span a broad range of binding site strengths (Fig. 1c)[13]. We used the synthetic $O_{sym}$ operator as well as the naturally occurring $O_1$ and $O_2$ operators as starting points, and constructed mutants of each operator (Supplementary Table 2). These promoter-operator constructs were used to drive the expression of the fluorescent protein mVenusNB (Fig. 1c).

For each operator, we quantified gene expression using single-cell mRNA FISH (Fig. 1c, "mRNA", Supplementary Fig. 2), allowing us to determine the distribution of mRNA copy numbers across a population of fixed cells. As expected, the wide range of binding site strengths was reflected in mean mRNA copy number values ranging from 0.3 to 20 (Supplementary Fig. 3). In parallel, we measured mVenusNB fluorescence for each construct (Fig. 1c, "mVenusNB") and found that mean mRNA and mean protein fluorescence were highly correlated across our panel of LacI binding sites, as expected (Fig. 1d). We also validated the mRNA FISH data with relative RT-qPCR measurements for a subset of binding sites, and found excellent agreement between the measurements (Supplementary Fig. 4).

### Calculation of $k_a$ and $k_d$ from mRNA statistics

In Fig. 2, the procedure for the calculation of $k_a$ and $k_d$ is illustrated using the $O_{sym}$ operator as an example. Equations (1) and (2) contain two unknowns besides $k_a$ and $k_d$: the degradation rate $\gamma$ and the basal transcription rate $r$. The degradation rate $\gamma$ was obtained for the $O_{sym}$, $O_1$, and $O_2$ operators by halting transcription at $t = 0$ using rifampicin (500 μg/mL) and measuring mRNA levels at subsequent time points

using bulk RT-qPCR, yielding a value of $\gamma = 0.79$ min$^{-1}$ (Fig. 2a, left), consistent with previously published results[16,17]. The basal transcription rate $r$ was determined separately for each operator by measuring the mean mRNA copy number in Δ*lacI* strains and using the fact that $\mu_{\Delta lacI} = r/\gamma$, enabling estimation of $r$ once the degradation rate $\gamma$ and mean expression $\mu_{\Delta lacI}$ are known (Fig. 2a, right). For the $O_{sym}$ operator, we estimated that $r = 24.6$ min$^{-1}$. With $r$ and $\gamma$ known, and the mean and Fano factor experimentally determined using mRNA FISH, Eqs. (1) and (2) constitute a system of two equations with two unknowns and can be numerically solved for $k_a$ and $k_d$ (Supplementary Fig. 5). Before doing so, we correct the Fano factor for the effect of RNAP copy number variability to obtain the corrected Fano factor $F_c = F - \mu/10$, as described previously[18,19], then use $F_c$ in place of F when solving for $k_a$ and $k_d$ (Fig. 2b). The inferred $k_a$ and $k_d$ values are substantively unchanged even if this correction is not included (Supplementary Fig. 6). For both Δ*lacI* and *wt lacI* experiments, only cells with lengths between 1.73 and 2.16 μm were included in the analysis (out of a length range of ~1.4–3.25 μm across all cells), to ensure that only cells with a single copy of the promoter-operator construct were present and thus counteract variability from gene copy number variation[18,19].

This procedure was repeated to estimate $k_a$ and $k_d$ for each of the 35 LacI binding sites. The value of $r$ was estimated individually for each operator from Δ*lacI* experiments, in order to account for potential differences in basal transcription rates due to changes in operator sequence, which in turn affects the 5′ UTR and hence potentially transcription initiation. We found that changes in operator sequence weakly affected transcription in Δ*lacI* strains, with a maximum difference of about twofold (Supplementary Fig. 7). In Fig. 3a, the experimentally-determined mRNA copy number distributions are shown for the $O_{sym}$, $O_1$, and $O_2$ operators (colored histograms), along with corresponding model predictions (see section "Methods", "Analytic probability mass function") given the estimated values of $k_a$ and $k_d$ (black dotted lines), which show good agreement. The corresponding distributions are shown in Supplementary Fig. 3 for all operators.

In Fig. 3b, the estimated $k_a$ values are plotted against $k_d$ for all operators, revealing a distinct anticorrelation between these two rates, and demonstrating that changes in operator strength are reflected in both rates, rather than primarily in $k_d$. $O_{sym}$ mutants contribute most to the anticorrelation, presumably because $O_{sym}$ is the strongest operator and its mutants contain the most information about LacI kinetics in their mRNA distributions. In Supplementary Fig. 8, we compare our results with previous in vitro measurements of the same operators[4] and find reasonable agreement between in vitro operator occupancy and in vivo repression ratio (Supplementary Fig. 8a), as well as between in vitro and in vivo rate measurements (Supplementary Fig. 8b, c).

### Discussion

Here, we combined a mathematical model of stochastic gene expression with measurements of steady-state mRNA copy number distributions, to infer the LacI association and dissociation rates for a wide range of operator strengths. This system can be described with more complex models explicitly incorporating, e.g., RNAP binding kinetics, open complex formation, and transcription-induced supercoiling[7,20–23], but our aim was to employ the simplest model that adequately described the observed distributions. For some operators with higher mean expression, the theoretical probability mass function (pmf) fits the data less well, which presumably reflects the fact that our approach is less suitable for weak operators whose mRNA distributions contain less information about LacI binding kinetics. We also see that several operators deviate from the theoretical pmf in the 0 and 1 mRNA bins, which may be due to the presence of partially transcribed or decayed mRNAs.

Nonetheless, we found good agreement between our measurements and previous in vivo results, strengthening the credibility of our

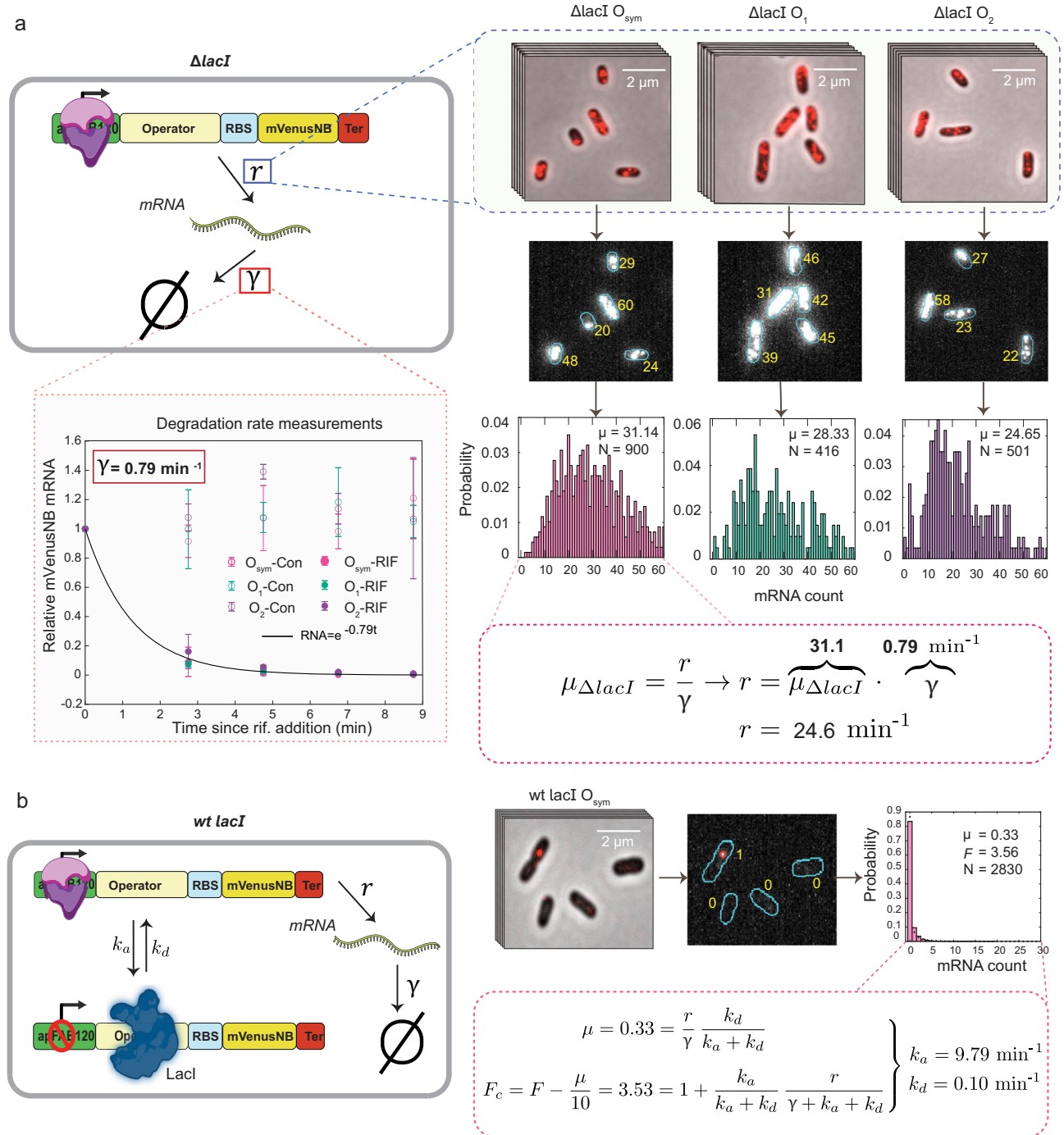

**Fig. 2 | Calculation of $k_a$ and $k_d$ for $O_{sym}$ operator. a** First, $r$ and $\gamma$ are determined in a strain in which LacI is deleted (created in BioRender https://BioRender.com/q45e576). $\gamma$ is determined by performing bulk RT-qPCR on the *mVenusNB* mRNA on samples taken at 2-min intervals following addition of rifampicin to halt transcription. Data points labeled "RIF" correspond to rifampicin-treated samples while points labeled "con" represent untreated samples. Error bars represent standard deviation of three biological replicates. $r$ is determined by measuring mean mRNA expression $\mu_{\Delta lacI}$ in $\Delta lacI$ strains using mRNA FISH, and using the equation $\mu_{\Delta lacI} = r/\gamma$. **b** Once $r$ and $\gamma$ are known, $k_a$ and $k_d$ are determined by measuring the mRNA copy number distribution in cells expressing LacI (created in BioRender https://BioRender.com/q45e576). From the distribution, the mean $\mu$ and Fano factor $F$ are computed. The Fano factor is corrected for the effect of RNAP copy number variation (see main text) to obtain the corrected Fano factor $F_c$, whereupon the appropriate values for $\mu$, $F_c$, $r$, and $\gamma$ are substituted in Eqs. (1) and (2), which are solved numerically for $k_a$ and $k_d$. Source data are provided as a Source Data file.

approach. For the $O_{sym}$ operator, we obtained a dissociation rate of $k_d = 0.10$ min⁻¹, in excellent agreement with $k_d = 0.11$ min⁻¹ as obtained in a previous in vivo study via microscopy on fluorescently-tagged LacI[24]. For the $O_1$ operator, we found an association rate of $k_a = 5.23$ min⁻¹, slightly higher than $k_a \cong 2.5$ min⁻¹ obtained in a previous study (after accounting for LacI expression levels)[25]. We attribute the

twofold higher association rate in our work to the fact that we used the *wt* LacI protein which tetramerizes in vivo, whereas the previous study used a fluorescent LacI-venus fusion which dimerizes but is incapable of tetramerization and intersegmental transfer. To our knowledge, our work represents the first in vivo measurement of search times for the *wt* LacI protein.

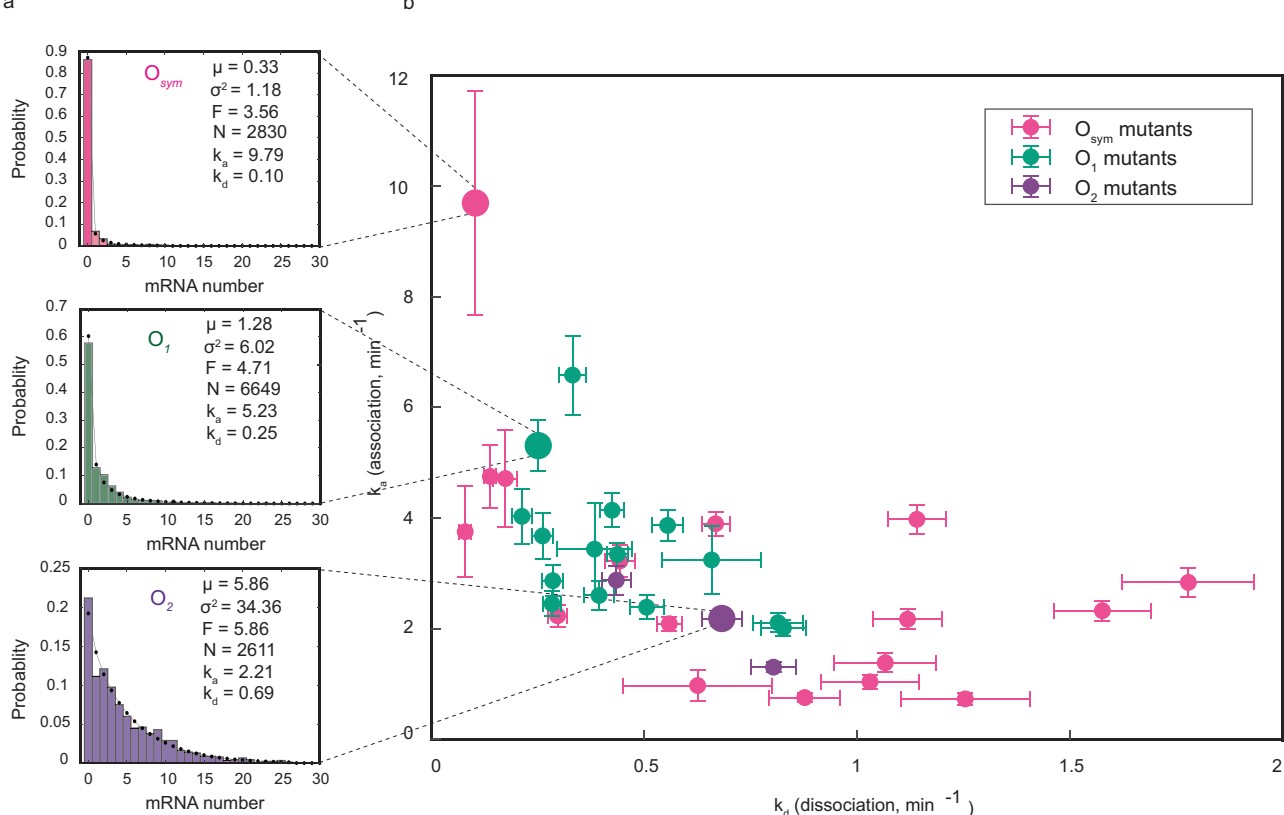

**Fig. 3 | Selected mRNA distributions and inferred rates for all operators.**
**a** Experimentally observed mRNA copy number distributions are shown for the
$O_{sym}$, $O_1$, and $O_2$ operators, along with the model prediction given inferred rate
parameters (black dots). **b** Association rate plotted against dissociation rate for all
35 operators assayed, showing a pronounced anticorrelation. Error bars represent
SEM obtained by bootstrapping. Source data are provided as a Source Data file.

Our measurements also show qualitative agreement with previous
in vitro results[13] in that equilibrium occupancy, $k_a$, and $k_d$ values are
generally correlated in vivo vs. in vitro (Supplementary Fig. 8). How-
ever, both $k_a$, and $k_d$ values are generally about an order of magnitude
larger in vivo. Several important differences between in vivo and
in vitro contexts make exact agreement unlikely; these include LacI
concentration, salt concentration[26], presence of non-specific DNA and
other DNA-binding proteins, and DNA topology (summarized in Sup-
plementary Table 1). While the faster dissociation rate in vivo can
seemingly be explained by the higher salt concentration (which
impedes sliding and lowers the probability of rebinding), the majority
of factors identified could be expected to result in slower association
in vivo. For this reason, it seems noteworthy that we actually observe
faster association in vivo, evidently reflecting LacI's impressive opti-
mization for fast search in living cells. Interestingly, the synthetic $O_{sym}$
variants appear to exhibit relatively lower $k_a$ values and higher $k_d$
values in vivo than $O_1$ and $O_2$ variants with similar in vitro rates (Sup-
plementary Fig. 8); we are uncertain as to why.

This paper provides direct in vivo evidence of an anticorrelation
between TF association and dissociation rates. Such anticorrelation
appears if binding strength mainly is governed by the probability of
recognizing and binding upon reaching the operator, which also shows
up in the macroscopic rate of leaving the operator $k_d$ since many
rebinding events occur for strong operators[13,27]. For this reason, it
may be a general feature of DNA-binding proteins whose search
mechanisms combine 1D sliding and 3D diffusion. This picture is
contrary to the long-standing hypotheses that TF binding site
strength is primarily dictated by the dissociation rate, but has the
evolutionary advantage that TFs do not get stuck in near-operator
sequences while searching.

At the same time, finding an appropriate regime in parameter
space for basal transcription rate $r$, repressor concentration, and
operator strength in order to carry out these measurements was not
trivial. For strong operators such as $O_{sym}$, $r$ needed to be large enough
compared to the association rate $k_a$ for multiple transcripts to have a
reasonable chance of being produced during LacI unbinding events.
Otherwise, if each LacI unbinding event leads to 0 or 1 transcripts,
transcription events become uncorrelated and hence effectively
Poissonian, with the Fano factor equal to one. In this scenario, the
variability carries no information about LacI kinetics. Conversely, too
large an $r$ value risks perturbing LacI binding kinetics as RNAP clears
the promoter region, potentially hindering LacI from binding[21]. In
Supplementary Fig. 9, we investigate this effect and find that it does
not materially affect the conclusions of this paper. It was also necessary
to balance LacI concentration with operator strength such that
operators are neither always bound nor never bound, situations in
which gene expression variability again carries little information about
LacI kinetics (Supplementary Fig. 10). As the operators investigated
here roughly spanned the range of operator strengths found in nature
(from stronger than $O_1$ to weaker than $O_2$), the wild-type LacI con-
centration (ca 5–10 tetramers per cell[28]) proved to be an appropriate
choice.

In other words, we sought to create a scenario with a simple
regulatory architecture where parameter values were chosen to make
the connection between cell-to-cell variability and LacI kinetics as clear
as possible. Nevertheless, these parameters generally fall within the
range observed in nature, and the promoter architecture studied here
("simple repression") is one of the most common in *E. coli*[29]. For more
complicated regulatory architectures, TF kinetics remain an important
factor dictating both mean and variability in gene expression. One

implication of our work is that TF association and dissociation rates cannot be tuned independently by mutations in binding sites. This in turn should in principle constrain the changes in gene expression output that can be accessed by mutations in regulatory DNA, whether by evolution in nature or intentional design in a synthetic biology context. However, unambiguously detecting the signature of such constraints may be challenging given the multitude of factors that affect gene expression. A complete understanding of the interplay between TF binding site strength, TF kinetics, gene expression variability, and the evolution of transcriptional regulatory sequences remains a compelling challenge for ongoing research.

## Methods

### Strain construction

Chromosomal knockout of *lacI* gene: the *lacI* gene was deleted using the DIRex method[30]. PCR fragments were amplified from the Acatsac1 (GenBank: MF124798) and Acatsac3 cassettes (GenBank: MF124799) using the oligos Del_LacI_Direx-F cat_mid-R, cat_mid-F, and Del_LacI_Direx-R (see Supplementary Table 3 for sequences). The PCR products were purified and electroporated into EL330 [*E. coli* K-12 MG1655] cells expressing lambda red proteins from plasmid pSIM5-Tet. After electroporation, cells were incubated in 1 mL of SOC medium at 30 °C for 1 h and then plated onto LB + 25 µg/mL chloramphenicol plates. The colonies were re-streaked on LB + 5% sucrose plates to select cells with the desired recombination. The resulting colonies were confirmed by PCR. Further sequence confirmation was done for the deletion of LacI gene in the chromosome and hereafter the strain is referred to as EL4110 [*E. coli* K-12 MG1655, *ΔlacI*].

### Operator mutation library construction

Using Golden Gate cloning, we assembled a plasmid with an R6K origin of replication backbone and a Biopart that consists of the fluorescent protein (mVenusNB) with a strong ribosome binding site downstream of the *lac* operator site, with expression driven by the constitutive synthetic promoter apFAB120[31]. Upstream of the promoter sequence, an ampicillin resistance cassette was introduced transcribing in the opposite direction (Supplementary Fig. 1).

Next, we amplified the Biopart with *ygaY* homology arms using the oligos ygaY_HA_F, and ygaY_HA_R, in preparation for chromosomal integration into the *ygaY* locus. The PCR fragment was further purified and electroporated into the EL330 and EL4110 strains producing lambda red proteins from plasmid pSIM5-tet. The successful chromosomal integration of the Biopart in the *ygaY* locus was confirmed by PCR and sequencing. For the operator mutation library, we selected single- and double-point mutations in $O_{sym}$, $O_1$, and $O_2$ from mutants characterized in Marklund et al.[13]. Using the DIRex method, we constructed a total of 35 operator variants (see Supplementary Table 2) in both the EL330 and EL4110 strains. All these strains were also confirmed by sequencing.

### Growth medium and conditions

From the glycerol stocks (−80 °C), cells were streaked on the fresh LB plates and grown overnight at 37 °C. Single colonies were inoculated in an LB medium and grown overnight in a continuous orbital shaking incubator with 200 rpm at 30 °C. From the overnight cultures, cells were further diluted in 1: 100 times in fresh M9 medium (containing M9 salts, 2 mM MgSO4, 0.1 mM CaCl2, 0.4% succinate as a carbon source, supplemented with 0.5X RPMI). Cells were grown at 37 °C with 200 rpm until they reached mid-exponential phase ($OD_{600}$ = -0.4). Next, cells were collected for microscopy experiments to quantify mRNA and protein expression levels.

### FISH probe design

To detect the mRNA of *mVenusNB* gene, 30 fluorescent DNA probes were designed using the Stellaris® Probe Designer version 4.2 (https://

www.biosearchtech.com/stellaris-designer) and purchased from Integrated DNA Technologies (IDT Iowa, USA). Each DNA probe was 20 nt in length with a minimum distance of 2 nt and the masking level at 1–2. All probes were labeled by Cy5 at the 3′ end of the DNA. Probe sequences are listed in Supplementary Table 4.

### FISH protocol

We followed the FISH protocol as in refs. 16,32. Briefly, when cell cultures reached mid-exponential phase ($OD_{600} \cong 0.4$), cells were collected by centrifugation at $2400 \times g$ for 5 min. Pelleted cells were then fixed by resuspending them gently in 1 mL of 3.7% (v/v) formaldehyde in 1× PBS prepared in nuclease-free water, followed by 30 min incubation at room temperature. Next, cells were centrifuged and washed twice in 1 mL of 1× PBS, followed by permeabilization in 1 mL of 70% ethanol (v/v) in nuclease-free water, with gentle mixing at room temperature for 1 h. Afterward, they were washed once more in 1 mL of washing buffer (2× SSC solution in nuclease-free water containing 40% (w/v) formamide). Probing was done by treating the cells with hybridization buffer (consisting of 2× SSC, 40% (w/v) formamide, 10% (w/v) dextran sulfate, 2 mM ribonucleoside-vanadyl complex, 0.2 mg/mL BSA, and 1 mg/mL carrier *E. coli* tRNA, along with 1 µM of each fluorescent probe), overnight at 30 °C. Next, the cells were washed thrice with the washing buffer to remove excess probes, and subsequently resuspended in 2× SSC buffer. 3 µL of the cell suspension were sandwiched between the coverslip and 1% (w/v) agarose pads. In order to improve the photostability of Cy5 dye, the agarose pads were enriched with an oxygen scavenging system[33] consisting of 2.5 mM protocatechuic acid (Sigma, prepared from a 100 mM stock stored frozen in water-NaOH, pH 8) and 0.05 U/mL protocatechuate 3,4-dioxygenase (OYC Americas).

### RT-qPCR measurements of *mVenusNB* expression

As in FISH experiments, overnight cultures were diluted 1:100 into fresh M9 medium and grown until mid-exponential phase. Next, cultures were treated with twice the volume of RNAprotect Bacteria reagent (Qiagen, Germany) at room temperature for 5 min and centrifuged at $2400 \times g$ for 10 min. An enzymatic lysis was performed on the pelleted cells with Lysozyme (10 mg/mL) in Tris-EDTA buffer (pH 8.0) and 10 % SDS. From these lysates, the total RNA content was extracted using the PureLink RNA Mini Kit (ThermoFisher) as per the kit manufacturer's instructions. The RNA content and absorbance ratios A260/A280 nm and A260/A230 nm were quantified by a Nanodrop 2000 Spectrophotometer (Thermo Scientific). The ratio (2.0–2.1) indicated highly purified RNA.

DNA contamination was removed by treating the samples with DNase using the Turbo DNA-free kit (Invitrogen). Next, cDNA synthesis was performed from RNA through the High Capacity Reverse Transcription kit (ThermoFisher) as per the manufacturer's instructions.

cDNA samples were mixed with qPCR master mix with the Power SYBR Green PCR mix (ThermoFisher) with primers (200 nM) for the target and reference gene. The primer sequences for the target (*mVenusNB*) and reference (*rrsA*) genes were shown in Supplementary Table 3. Experiments were performed in a StepOne™ Real-Time PCR system v2.3 (ThermoFisher). The thermal cycling conditions were 50 °C for 10 min, 95 °C for 2 min, followed by 40 cycles of 95 °C for 15 s, 60 °C for 30 s, 72 °C for 40 s with the fluorescence being read after each cycle, and finally a melting curve from 90 °C through 70 °C at 0.3 °C intervals and 1 s dwell time. All samples were performed in three technical replicates, and for each condition, No-reverse-transcriptase and no-template controls were used to crosscheck non-specific signals and contamination. qPCR efficiencies of these reactions were greater than 95%. The fold change was calculated from the $C_T$ values from target gene (normalized to the reference gene) and standard error, using Livak's $2^{-\Delta\Delta CT}$ method[34].

## mRNA degradation rates

To measure the *mVenusNB* mRNA lifetimes in strains varying the operator sites ($O_{sym}$, $O_1$, and $O_2$), we followed the procedure described in ref. [16]. From overnight cultures, cells were diluted 1:100 in a 20 mL volume of M9 succinate medium supplemented with 0.5X RPMI, grown at 37 °C with 200 rpm. Upon reaching $OD_{600} \cong 0.4$, the culture was divided into two equal halves, with each half transferred to a new 50 mL tube and kept at 37 °C with aeration. Subsequently, 1.0 mL of culture was extracted from each tube and mixed with 3 mL of RNA-protect Bacteria reagent (Qiagen, Germany) to stabilize cellular RNA, serving as the $t = 0$ samples. Next, to inhibit transcription a final concentration of 500 µg/mL of Rifampicin was added to one of the culture tubes, while the culture without rifampicin served as a control. At 2-min intervals after rifampicin addition (e.g., at $t = 2, 4, 6, 8$ min), 1.0 mL of culture was extracted from each tube and mixed with 3 mL of Qiagen RNAprotect Bacterial reagent. Subsequent total RNA extraction and quantitative real-time polymerase chain reaction were carried out following the procedures outlined above. The relative *mVenusNB* mRNA levels in the rifampicin-treated sample were fitted to an exponential function, $RNA = \exp(-\gamma \cdot t)$ where $\gamma$ is the mRNA degradation rate and $t$ is the time since rifampicin addition, yielding an estimate of $\gamma = 0.79 \, min^{-1}$.

## Microscopy

Wide-field microscopy was employed to capture cells in both phase contrast and fluorescent channels. The optical setup consisted of a Nikon Ti2-E inverted microscope equipped with a 1.45/100x oil immersion objective lens (CFI Plan APO lambda, Nikon), a Spectra III light source from Lumencor for epi-fluorescence illumination, and a Kinetix camera from Teledyne Photometrix. Control of the setup was facilitated by micro-Manager 1.4 software[35]. For RNA measurements, we acquired Z-stacks of 9 Cy5 fluorescence images centered on the focal plane and separated by 200 nm, with a filter cube consisting of an FF660-Di02 excitation filter and a Semrock 692/40 emission filter. A single-phase contrast image was captured at the focal plane. All images were captured at an exposure time of 250 ms. For protein measurements, we used the same optical setup, and fluorescence images were captured using the mVenusNB channel with a filter cube consisting of an FF01-559/34 (Semrock) excitation filter, a T585lpxr (Chroma) dichroic mirror, and a T590LP (Chroma) emission filter. We acquired images at 200 ms exposure for both phase and fluorescence images.

## Image analysis

The image analysis pipeline consists of the following steps: cell segmentation, spot detection, and mRNA or mVenusNB quantification (see below for mVenusNB quantification) using custom Matlab (R2022a) code. Cell segmentation was performed using the U-Net algorithm[36]. The segmentation was refined by imposing filters on cell area, width, and length; and each field of view was manually checked for mis-segmented cells. Fields of view with mis-segmented cells were excluded from further analysis.

For mRNA FISH experiments performed with *wt* LacI expression levels, spot detection proceeded similarly to previously published work[18]. Fluorescence images were subjected to a mild Gaussian filter ($\sigma_{XY} = 1$ pixel, $\sigma_Z = 0.7$ pixels), and local maxima were identified in 3D using Matlab's imregionalmax function. Local maxima whose peak intensities fell under a threshold were discarded. This threshold was chosen such that at most a handful of spots were detected in the negative control sample (a negative control sample was run for each individual experiment). For each detected spot whose intensity passed the threshold, the spot intensity was determined by summing the pixels inside a circle of radius 5 pixels of the local maximum, in the XY plane in which the maximum was detected. To avoid double-counting, spots whose quantification radii overlapped in a particular XY plane

were merged. The negative control pixel intensity was subtracted from all pixels while quantifying spot intensity.

To convert spot intensity to number of mRNA, the strain containing *wt lacI* and the $O_{sym}$ operator was used. This strain is highly repressed, with on average 0.33 mRNA per cell, such that most detected spots are likely to contain a single mRNA. A histogram of detected spot intensities was created for the $O_{sym}$ strain, and the single mRNA intensity was identified as the mode of the resulting intensity distribution (Supplementary Fig. 2).

With the single mRNA intensity in hand, the number of mRNA in each cell can be computed. The intensities from all detected spots in a cell are summed, and the resulting sum is divided by the single mRNA intensity, yielding the estimated number of mRNA in the cell. From the set of cells for each operator, the mean, Fano factor, and other statistics can be calculated. mRNA distributions in Figs. 2 and 3 were created by pooling the results of at least two independent experiments on separate days. Bootstrapped estimates of uncertainty in $k_a$ and $k_d$ (Fig. 3) were created as follows: for each operator, 1000 bootstrap samples were drawn from the pooled set of mRNA copy number observations. For each bootstrap sample, the $k_a$ and $k_d$ values were estimated as described above ("Calculation of $k_a$ and $k_d$ from mRNA statistics"). For error bars shown in Fig. 3b, the lower and upper error bar ranges represent the 10th and 90th percentile of bootstrapped $k_a$ and $k_d$ values, respectively.

For mRNA FISH experiments performed in $\Delta lacI$ strains, the summed fluorescence intensity of all pixels was calculated for each cell at the XY plane in which the cell was brightest, background fluorescence was subtracted using the negative control strain, and the background-subtracted summed fluorescence intensity was divided by the single mRNA intensity to yield the number of mRNA in each cell.

For mVenusNB expression experiments (Fig. 1c, d), the average mVenusNB channel pixel intensity was computed for each cell, and the average fluorescence of the negative control strain (without mVenusNB) was subtracted from this value. The background-subtracted mVenusNB intensity was averaged over all cells with a given operator to obtain a mean mVenusNB expression level for each operator. Error bars in Fig. 1d represent sem derived from pooling all cells with a given operator across experiments, drawing 10,000 bootstrap samples from the pooled data set, and computing the standard deviation of the bootstrapped mean values.

## Stochastic simulations

Stochastic simulations in Fig. 1b were performed using code from Sanchez et al.[37]. The steady-state probability distributions in Fig. 1b were computed from the chemical master equation by formulating the master equation in matrix form and numerically computing the eigenvectors of the matrix with eigenvalue equal to zero, also as described in Sanchez et al.

To generate Supplementary Fig. 9, investigating the effects of RNAP occlusion of the operator, this code was modified to make the association rate $k_a$ dependent on the time since the most recent transcript production event. Specifically, the association rate was set to zero for a time $t_{occlude}$ following each transcription event; after $t_{occlude}$, $k_a$ returned to its previously set value. Thus, $t_{occlude}$ models occlusion of the operator during transcription initiation, during which time the operator is unavailable for binding. When constructing Supplementary Fig. 9, the association rate $k_{a,sim}$ was modified for each operator such that the effective association rate when $t_{occlude} = 1$ s would approximately equal the experimentally measured $k_{a,exp}$ for that operator. For instance, in the case of $O_{sym}$, $r = 24.6 \, min^{-1}$ (Fig. 2) and $k_{a,exp} = 9.79 \, min^{-1}$. This means that, on average, 24.6 s of every minute are unavailable for LacI binding, so that the "true" association rate would be multiplied by a factor of $(60 − 24.6)/60 = 0.59$. Thus, to create the $O_{sym}$ simulations in Supplementary Fig. 9, $k_{a,sim}$ was set to

$k_{a,sim}$ = 9.79/0.59 min$^{-1}$ = 16.6 min$^{-1}$, which is indeed the inferred value for $k_a$ when $t_{occlude}$ = 0. An analogous procedure was used to determine $k_{a,sim}$ values for $O_1$ and $O_2$ that, when taking into account an occlusion time $t_{occlude}$ = 1 s, would yield inferred values close to the experimentally observed rates $k_{a,exp}$ and $k_{d,exp}$. For all three operators, this procedure worked well as inferred $k_a$ and $k_d$ values were close to the experimentally-determined values when $t_{occlude}$ = 1 s.

To generate Supplementary Fig. 10, a set of 50 simulated operators was created whose association rate per repressor $k_a^o$ and dissociation rate $k_d$ values were linearly spaced along the line defined in Marklund et al. (Supplementary Fig. 10, green lines). In order to capture the effect of varying repressor copy number $n_{repressor}$, the association rate $k_a$ was assumed to be proportional to repressor copy number: $k_a = n_{repressor} k_a^o$. We examined six different repressor copy numbers, varying from 1 to 100 per cell. For a given operator and repressor copy number, the steady-state mRNA copy number distribution was computed numerically, again as described in Sanchez et al.[37], and a set of 1000 samples (i.e. comparable to performing mRNA FISH on a population of 1000 cells) was drawn from the distribution. The observed mRNA copy numbers in the simulated population were then analyzed according to the procedure in Fig. 2, yielding the "Inferred values" (blue dots). In order to estimate the effects of experimental noise, each mRNA was assigned a "fluorescence intensity" randomly chosen from a Gaussian distribution with mean 1 and standard deviation 0.4. The intensities from all mRNA within a particular cell were summed and rounded to the nearest integer, similar to how experimental data are analyzed. These data were also analyzed according to the procedure in Fig. 2 ("Inferred values w exp noise", orange dots). Basal transcription rate $r$ = 22 min$^{-1}$ and mRNA degradation rate $\gamma$ = 0.80 min$^{-1}$ in these simulations.

## Analytic probability mass function

The analytic probability mass function plotted in Fig. 3a and Supplementary Fig. 3 was originally derived by Shahrezaei and Swain and later adapted by Morrison et al.[7,14].

For convenience and clarity, we reproduce it here. We start by defining non-dimensionalized versions of the various rates by dividing by the mRNA degradation rate, such that

$$\widetilde{r} = r/\gamma; \ \widetilde{k_a} = k_a/\gamma; \ \widetilde{k_d} = k_d/\gamma$$

The probability of a cell having $m$ mRNA is then given by

$$p(m) = \frac{\Gamma(\alpha+m)\Gamma(\beta+m)\Gamma\left(\widetilde{k_a}+\widetilde{k_d}\right)}{\Gamma(\alpha)\Gamma(\beta)\Gamma\left(\widetilde{k_a}+\widetilde{k_d}+m\right)} \frac{1}{m!} {}_2F_1\left(a+m, \beta+m, \widetilde{k_a}+\widetilde{k_d}+m, -1\right)$$

(3)

where

$$\alpha = \frac{1}{2}\left(\widetilde{r}+\widetilde{k_a}+\widetilde{k_d}+\sqrt{\left(\widetilde{r}+\widetilde{k_a}+\widetilde{k_d}\right)^2 - 4\widetilde{r}\widetilde{k_d}}\right)$$

$$\beta = \frac{1}{2}\left(\widetilde{r}+\widetilde{k_a}+\widetilde{k_d}-\sqrt{\left(\widetilde{r}+\widetilde{k_a}+\widetilde{k_d}\right)^2 - 4\widetilde{r}\widetilde{k_d}}\right),$$

${}_2F_1$ is the Gaussian hypergeometric function, and $\Gamma$ denotes the gamma function (i.e., the generalization of the factorial function, not to be confused with the mRNA degradation rate).

It is noteworthy that Eq. (3) depends only on the non-dimensionalized rates, as is also the case for Eqs. (1) and (2) as can be seen with minor rearranging. In other words, the absolute rates cannot be obtained from the mRNA data; what we are actually doing is using the mRNA data to solve for the non-dimensionalized rates, then using the external measurement of mRNA degradation rate $\gamma$ to convert to physical units of min$^{-1}$.

## Reporting summary

Further information on research design is available in the Nature Portfolio Reporting Summary linked to this article.

## Data availability

All experimental data are available in the SciLIfeLab Data Repository at https://doi.org/10.17044/scilifelab.26425576.v1[38]. Source data are provided with this paper.

## Code availability

All code used in this paper is available in the SciLifeLab Data Repository at https://doi.org/10.17044/scilifelab.26425576.v1[38].

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

## Acknowledgements

We thank Johan Paulsson, Jinwen Yuan, Emil Marklund, Gerrit Brandis, and Daniel Camsund for helpful discussions. We also thank Irmeli Barkefors for critical reading of the manuscript. D.J. acknowledges support from the Swedish Research Council (grant number 2020-05137). J.E. acknowledges funding from the ERC (advanced grant no. 885360), the Swedish Research Council (grant nos. 2016-06213 and 2018-03958), the Knut and Alice Wallenberg Foundation (grant nos. 2016.0077, 2017.0291, and 2019.0439), and the eSSENCE e-science initiative. Computation and data management were enabled by resources provided by the Swedish National Infrastructure for Computing at UPPMAX, partially funded by the Swedish Research Council through grant agreement no. 2018-05973.

## Author contributions

J.E. and D.J. conceived and supervised the project. V.K. performed experiments. V.K., S.Z., and D.J. analyzed data. D.J. performed simulations. V.K., J.E., and D.J. wrote the manuscript.

## Funding

## Competing interests

The authors declare no competing interests.
