## [Transparent Peer Review file · Nature Communications]

Anti-correlation of LacI association and dissociation rates observed in living cells

Corresponding Author: Dr Daniel Jones

Version 0:

Reviewer comments:

Reviewer #1

(Remarks to the Author)

For decades, the assumption underlying the understanding of TF dynamics has been that binding rates are dictated by diffusion while unbinding rates are dictated by electrostatic interactions between binding sites and TFs. In this manuscript, Kandavalli et al demonstrate that this assumption does not hold up. While this has been shown before in vitro, it is to my knowledge the first direct measurement in vivo.

The idea and presentation is very simple and thus very clear and I don't think there's a lot to criticize. My only real complain is that I could not find justification for the "corrected fano factor" to not be sufficient. Why is there a linear term that scales with mean? Why is the coefficient 1/10? Otherwise, there's some slight messiness with notation. For instance, there's references to fig. 3a and b, but there is no a or b in the figure and the math symbols are different in the text and figures (gamma in particular is distractingly distinct).

Otherwise I support publication of this work, I think it is useful, important and well done.

(Remarks on code availability)

Reviewer #2

(Remarks to the Author)

This manuscript explores the mechanistic origins of heterogeneity of gene expression in bacteria, focusing on the effect in the binding affinity of transcription factors to their cognate DNA sequences. The authors used strains carrying LacI and any of 35 operator variants of differing binding strengths controlling the expression of a fluorescent protein. They then measured the transcription activity by looking measuring RNA through Fluorescence In Situ Hybridization (FISH) microscopy, quantified the resulting protein from the fluorescence produced, and compared it to a mathematical model to obtain estimates of binding kinetics. Their results show an anti-correlation between association (k_a) and dissociation (k_d) rates, showing that the tightness of binding is a critical factor during target search. This is an important conclusion that corroborates previous work obtained by their group using in vitro approaches.

The manuscript is focused and concise, the authors present sufficient information to reach their conclusions, and the text is clear. The major issue identified in the work is the discrepancy for the estimates of k_a and k_d in vitro and in vivo (Fig. S7). There is a difference of about one order of magnitude between the two approaches. Although the ratios between k_a and k_d are in agreement, the implications for the kinetics of these proteins are very different, some of them fitting better with previous reports (dwell times and search times of LacI are reported to be in the minutes time scale). The authors only mention that their results agree with previous estimates, but they should explain where this discrepancy arises.

It would also be helpful for the authors to extend the discussion on the applicability model and implications of their model in the discussion section. Currently this topic is only briefly covered.

Other specific points:

- Figure 1b: Wrong colors are described in the figure legend. It should state: "For easier comparison, the 'fast' mRNA distribution is plotted (orange dashed line) on top of the 'slow' mRNA distribution (cyan bars), and vice versa."

- The text and figure legend refer to an A and B inset of Figure 3. However, no such division is found in the figure.

(Remarks on code availability)

Reviewer #3

(Remarks to the Author)

(Remarks on code availability)

Reviewer #4

(Remarks to the Author)

This paper investigated the sequence-specific binding kinetics of LacI in *E. coli* cells. The authors first established a mathematic model that is based on four unknown parameters, the mRNA degradation lifetime, the basal expression level of mRNA in the absence of LacI, and the association and dissociation rates (k_a and k_d) of LacI to lacO sequences. The authors determined the first two parameters experimentally by measuring mRNA decay rate when transcription is inhibited and by measuring mRNA copy number distribution using single-molecule FISH (smFISH) in a LacI strain. The authors then measured the mean and Fano factor of mRNA copy number distributions in cells containing different lacO variants in the presence of LacI expression to determine the remaining two parameters, k_a and k_{off} for each lacO sequence. The work is an extension of the group's early work on the in vitro binding measurement of LacI and reached the similar conclusion that k_a and k_d are anticorrelated in vivo. This work nicely integrates theoretical modeling and elegant experimental designs to resolve the binding kinetics of LacI to a wide range of binding sequences in cells and provides a new framework to study transcription factor binding site search in a realistic cellular environment. We support the work's acceptance for publish provided that the authors address our major concerns below.

Major Concerns

1. The authors determined the k_a and k_d by solving the two equations of μ and F . The distribution of mRNA copy numbers contains these two parameters, and also the binding mRNA decay time and the basal expression level. Could an analytic solution for the mRNA copy number distribution be developed and the fit parameters be compared with those determined by the two equations? In Supplemental Fig. 3 the authors showed the distributions and the corresponding probability mass function overlay, but there is no description of the probability mass function (pmf) in the text. Is the function derived by numerical simulation or by solving the analytical form of the model? Some of the pmf do not appear to describe the distribution well, especially the ones with O_{sym} . Also, most of the distributions deviated from pmf in the first 1-2 bins. Why?

In addition to including distribution-based fitting, it would be helpful to include more details about the robustness of the fitting for k_a and k_d . Possibly in the supplemental, can you explain if there are any parameter values that give you infinitely many solutions or no solutions? Additionally, how sensitive is the fitting to the experimentally measured parameters? For example, if γ is off by 0.1 min^{-1} , how much would k_a and k_d vary.

2. We are impressed by the level of kinetic detail obtained from the mRNA copy number distributions. If there are no technical limitations specific to this work, single molecule tracking could provide a more direct measurement of binding kinetics. Either the authors could compare their results to the kinetics measured in Elf et al 2007 (DOI: 10.1126/science.1141967) or could provide similar measurements for a few of the sequences tested in this work.

3. The authors discussed that the in vivo measurements of k_a and k_d in *E. coli* are largely correlated with in vitro measurements in their previous publication. However, supplemental Fig. 7 mainly showed that they are proportional (repression ratio in vivo vs. relative occupancy in vitro. Fig. S7a), but the absolute values of k_a and k_d in vivo were not really correlated with in vitro measurement, especially for O_{sym} (Fig. S7b, c). The discrepancy between the absolute values of k_a/k_d between in vivo and in vitro is expected because of the crowded cellular environment and the presence of large nonspecific binding sequences nearby. Please discuss the implications of these differences—

we find that the O_{sym} being a synthetic sequence being less convincing.

4. It is significant to validate in vivo the major conclusion from the authors' previous work: the binding rate is dependent on the binding sequence, implying that the actual binding is determined by microscopic interactions after the initial bimolecular collisions.

However, by this definition, shouldn't it be natural that k_a and k_d are anticorrelated since k_a includes the microscopic state interactions? Additionally, the anticorrelation plot (Fig. 3) is mainly defined by lacOsym—if data points from lacOsym were taken out, the anticorrelation is much less significant. But lacOsym is also the one that appeared most different from in vitro measurement. Could the authors please clarify these points?

(Remarks on code availability)

Reviewer #5

(Remarks to the Author)

(Remarks on code availability)

Version 1:

Reviewer comments:

Reviewer #1

(Remarks to the Author)

The authors have answered my concerns.

(Remarks on code availability)

Reviewer #2

(Remarks to the Author)

The authors have addressed all the previously raised comments.

(Remarks on code availability)

Reviewer #3

(Remarks to the Author)

(Remarks on code availability)

Didn't review the code

Reviewer #4

(Remarks to the Author)

We appreciate that the authors addressed most of our concerns in a much more thorough discussion section. Also, we find the addition of supplementary table 1 to be clear and concise and the addition of the analytic probability mass function to be helpful.

One minor concern is that whether the proportion of partially transcribed or degraded mRNAs would be indeed large enough to cause some of the very large 0 bins especially in the bottom row of supplementary figure 3. It is interesting to note that the 0 bin is overpopulated for the high expressing promoters, indicating that the pattern may not be due to stochastic burstiness. It is difficult to imagine such an association is specific for highly expressed promoters. Do the authors have any other explanations?

(Remarks on code availability)

Reviewer #5

(Remarks to the Author)

(Remarks on code availability)

Reviewer #1 (Remarks to the Author):

For decades, the assumption underlying the understanding of TF dynamics has been that binding rates are dictated by diffusion while unbinding rates are dictated by electrostatic interactions between binding sites and TFs. In this manuscript, Kandavalli et al demonstrate that this assumption does not hold up. While this has been shown before in vitro, it is to my knowledge the first direct measurement in vivo.

The idea and presentation is very simple and thus very clear and I don't think there's a lot to criticize. My only real complain is that I could not find justification for the "corrected fano factor" to not be sufficient. Why is there a linear term that scales with mean? Why is the coefficient 1/10?

This term arises from the estimated contribution from RNA polymerase copy number variation. The coefficient of 1/10 essentially arises from the fact that RNAP copy number variation has an estimated coefficient of variation of 1/10. We do appreciate the fact that this correction could seem somewhat ad hoc, but it was arrived at independently in two separate peer-reviewed publications from different groups, first in Jones, D. L. et al 2014, "Promoter architecture dictates cell-to-cell variability in gene expression" *Science* **346**, 1533–1536 (see equation S54 in the supplementary material), and second in J.R. Peterson, et al 2015, "Effects of DNA replication on mRNA noise" *Proc. Natl. Acad. Sci. U.S.A.* **112** (52) 15886-15891 (see equation 8).

Most importantly, as we show in Fig. S6, the main results of this paper remain the same even if this correction is not included.

Otherwise, there's some slight messiness with notation. For instance, there's references to fig. 3a and b, but there is no a or b in the figure and the math symbols are different in the text and figures (gamma in particular is distractingly distinct).

We thank the reviewer for pointing out these issues, which have now been fixed.

Otherwise I support publication of this work, I think it is useful, important and well done.

Reviewer #2 (Remarks to the Author):

This manuscript explores the mechanistic origins of heterogeneity of gene expression in bacteria, focusing on the effect in the binding affinity of transcription factors to their cognate DNA sequences. The authors used strains carrying LacI and any of 35 operator variants of differing binding strengths controlling the expression of a fluorescent protein. They then measured the transcription activity by looking measuring RNA through Fluorescence In Situ Hybridization (FISH) microscopy, quantified the resulting protein from the fluorescence produced, and compared it to a mathematical model to obtain estimates of binding kinetics. Their results show an anti-correlation between association (k_a) and dissociation (k_d) rates, showing that the tightness of binding is a critical factor during target search. This is an important conclusion that corroborates previous work obtained by their group using in vitro approaches.

The manuscript is focused and concise, the authors present sufficient information to reach their conclusions, and the text is clear. The major issue identified in the work is the discrepancy for the estimates of k_a and k_d *in vitro* and *in vivo* (Fig. S7). There is a difference of about one order of magnitude between the two approaches. Although the ratios between k_a and k_d are in agreement, the implications for the kinetics of these proteins are very different, some of them fitting better with previous reports (dwell times and search times of LacI are reported to be in the minutes time scale). The authors only mention that their results agree with previous estimates, but they should explain where this discrepancy arises.

There are a number of important differences between the *in vivo* and *in vitro* contexts that can substantively affect binding kinetics. One of the more easily-quantifiable is LacI concentration: in the *in vitro* data, LacI was flowed in at a concentration of 1.7 nM, whereas the *in vivo* LacI concentration is approximately 10 nM. As k_a is proportional to LacI concentration, the ~10-fold larger k_a *in vivo* seems to match the ~10-fold larger LacI concentration.

Another important difference is salt concentration. The *in vivo* concentration of monovalent cations (largely Na⁺ and K⁺) is 200-300 mM, whereas the corresponding concentration in *in vitro* experiments was 30 mM. In general, lower salt concentration is expected to yield enhanced sliding due to fewer positive ions masking electrostatic interactions between LacI and the DNA backbone. More sliding should in turn lead to a lower macroscopic dissociation rate as LacI has a higher probability of sliding back to the operator immediately after dissociating. This was demonstrated experimentally in Marklund *et al* where *in vitro* dissociation was measured for the O1 operator at salt concentrations ranging from 0 to 250 mM NaCl, and a ~ten-fold increase in k_d was observed over this range. Thus the *in vivo* / *in vitro* discrepancy in k_d could reasonably be explained by differing salt concentrations.

However, LacI and salt concentrations are far from the only relevant differences between these contexts. The crowded *in vivo* intracellular environment is considerably more complex, with high densities of cytoplasmic and DNA binding proteins, which can serve as “roadblocks” to LacI sliding, block LacI’s access to operators, induce DNA bending, and (as in the case of nucleoid associated proteins) alter DNA topology. The ratio of nonspecific to specific DNA is also vastly different, as *in vivo* the operator is part of a 5 million base pair genome, whereas *in vitro* the operator is part of several hundred base pair long DNA molecules on a microarray. These factors would be expected to slow LacI association, so it is interesting that we nonetheless see larger k_a values *in vivo*.

Finally, the LacI proteins themselves are not the same. The *in vitro* proteins are Cy3-labelled dimers with the tetramerization domain removed. Some cysteine residues were replaced to facilitate Cy3 labeling. These mutations were distal to the DNA binding domain, and Cy3 is a small organic dye, so the effect of cysteine replacement and labeling is predicted to be small. However, the effect of dimer vs. tetramer may be more significant and we estimate that the *wt* tetramer binds about twice as quickly as the dimer (see response to reviewer #4 point #2 below).

To sum up, there are numerous differences between *in vivo* and *in vitro* contexts, some of which (i.e. LacI and salt concentration) appear to be enough on their own to account for a 10-fold difference in kinetic rates. It is difficult to predict how these factors will interact when

combined together, but in any case, it would be very surprising if they somehow combined to yield no net differences. Thus we would only expect a relative rather than absolute correspondence between *in vivo* and *in vitro* kinetics, which is what we see. Finally, as discussed below, our *in vivo* measurements appear to be consistent with other *in vivo* measurements performed using different techniques.

We have added a discussion of these factors to the Discussion in the main text as well as a supplementary table (Supplementary Table 1).

It would also be helpful for the authors to extend the discussion on the applicability model and implications of their model in the discussion section. Currently this topic is only briefly covered.

We are not entirely sure whether the reviewer is referring to the model in terms of the random telegraph model of gene expression or the observed anticorrelation between association and dissociation rates (with the implication that binding strength is governed by the recognition probability), or both. Regarding the random telegraph model, we have added some discussion of potential model limitations in the context of incomplete agreement between the theoretical probability mass function and the observed mRNA distributions (see also response to reviewer #4 below).

Regarding the k_a - k_d anticorrelation and operator recognition probability: the insights derived from this study could also have broader implications beyond the specific case of LacI. For instance, the observed anticorrelation between k_a and k_d may be a general feature of DNA binding proteins that rely on a similar search strategy combining 1D (sliding) and 3D diffusion to locate their target sites. The anticorrelation could be seen as a constraint, as k_a and k_d can't be modified independently by changes in operator sequence. This could be relevant when attempting to e.g. design synthetic regulatory sequences to achieve targeted levels of cell-to-cell variability.

As described in the Discussion, in this work we sought to create a scenario with a relatively simple regulatory architecture with parameter values chosen such that the connection between cell-to-cell variability and LacI binding kinetics would be as clear as possible. Insofar as gene expression mean and variability from more complex promoter architectures are a function of TF association and dissociation kinetics, among other factors (modeled in e.g. Sanchez et al 2011 among others), we would certainly expect changes in gene expression distributions due to promoter region mutations to be materially affected by the observed relationship between association and dissociation rates, which we believe is likely to hold for other TFs than LacI. However, in more complicated scenarios, the number of possible transitions (with associated rates) between promoter states increases in principle as the square of the number of possible promoter states. In the case of activators, some activators act on the level of closed complex formation, while others act on open complex formation, which in turn entails more detailed modeling of RNAP-promoter kinetics. These and other factors make it probably quite difficult to infer kinetic rates from mRNA distributions (and thus to detect a clear signal of k_a/k_d anticorrelation) as we did here. Still, this would be an interesting challenge for future work.

We have expanded the Discussion to include these points while trying to maintain a reasonable level of concision.

Other specific points:

- Figure 1b: Wrong colors are described in the figure legend. It should state: "For easier comparison, the 'fast' mRNA distribution is plotted (orange dashed line) on top of the 'slow' mRNA distribution (cyan bars), and vice versa."

- The text and figure legend refer to an A and B inset of Figure 3. However, no such division is found in the figure.

We thank the reviewer for pointing out these issues, which have now been fixed.

Reviewer #3 (Remarks to the Author):

[Co-reviewer]

Reviewer #4 (Remarks to the Author):

This paper investigated the sequence-specific binding kinetics of LacI in *E. coli* cells. The authors first established a mathematic model that is based on four unknown parameters, the mRNA degradation lifetime, the basal expression level of mRNA in the absence of LacI, and the association and dissociation rates (k_a and k_d) of LacI to lacO sequences. The authors determined the first two parameters experimentally by measuring mRNA decay rate when transcription is inhibited and by measuring mRNA copy number distribution using single-molecule FISH (smFISH) in a Δ LacI strain. The authors then measured the mean and Fano factor of mRNA copy number distributions in cells containing different lacO variants in the presence of LacI expression to determine the remaining two parameters, k_a and k_{off} for each lacO sequence. The work is an extension of the group's early work on the in vitro binding measurement of LacI and reached the similar conclusion that k_a and k_d are anticorrelated in vivo. This work nicely integrates theoretical modeling and elegant experimental designs to resolve the binding kinetics of LacI to a wide range of binding sequences in cells and provides a new framework to study transcription factor binding site search in a realistic cellular environment. We support the work's acceptance for publish provided that the authors address our major concerns below.

Major Concerns

1. The authors determined the k_a and k_d by solving the two equations of m and F . The distribution of mRNA copy numbers contains these two parameters, and also the binding mRNA decay time and the basal expression level. Could an analytic solution for the mRNA copy number distribution be developed and the fit parameters be compared with those determined by the two equations?

Although an analytic solution has in fact been developed (see below), we have found it challenging to work with. The solution is based on the Gaussian hypergeometric function which does not have a closed-form solution. Instead, the hypergeometric function has to be evaluated by numerically integrating a second-order differential equation for each value at which one wants to evaluate the function. We found this to be very slow.

In any case, it's worth digging deeper into what fitting the analytic distribution would actually entail. We would interpret this to mean performing a maximum likelihood or Bayesian analysis to determine the likelihood/posterior probability of different k_a/k_d values given the observed mRNA counts data. In fact, a highly relevant paper to this question was published

recently (Morrison et al PLoS Comp Biol 2022). In this work, the authors examined exactly the scenario described here (stochastic mRNA expression from a repressed promoter, i.e. the “random telegraph” model) and developed code to perform Bayesian inference on the resulting pmf to determine k_a and k_d values. They used this code to re-analyze mRNA count data from Jones et al 2014 (which dealt with a similar promoter construct, but largely focused on changing repressor copy number rather than binding site strength).

This sounds highly relevant, so why didn't we simply reuse their code? First, on a practical level, we did download their code but had trouble getting it to run. More critically, the authors of Morrison et al communicated to us that they found it necessary to fix either k_a or k_d in order to get the analysis to converge, and for the analyses in their paper, they assumed that k_a is constant for different operators. As the main point of our paper was to interrogate the assumption of constant k_a , this would have defeated the purpose of our work. For this reason, we chose not to pursue this avenue further.

In summary, although we agree that fitting the distribution directly would be preferable in principle, the fact that our capable scientific colleagues recently devoted an entire paper to doing so, found it to be a non-trivial problem, and did not manage to create a solution that fit our needs, was enough to persuade us to use the approach that we took.

In Supplemental Fig. 3 the authors showed the distributions and the corresponding probability mass function overlay, but there is no description of the probability mass function (pmf) in the text. Is the function derived by numerical simulation or by solving the analytical form of the model?

The pmf plotted here is the analytical solution derived by Shahrezaei, V. & Swain, P. S. 2008 “Analytical distributions for stochastic gene expression”. *Proc. Natl. Acad. Sci.* **105**, 17256–1726 and later adapted by Morrison *et al* 2021. We have clarified this in the text, and have also reproduced the equation in the Methods section for reference.

Some of the pmf do not appear to describe the distribution well, especially the ones with O_{sym} .

It is true that some of the pmfs deviate appreciably from the experimental data, particularly for weaker operators with higher mean expression. We attribute this primarily to two factors: (1) the method we are using generally works less well for weak operators since intuitively the mRNA distribution contains less information about LacI binding kinetics (see for instance Supplementary Fig. S10 where analysis of simulated data starts to deviate significantly from simulation inputs); (2) although we attempted to control for factors such as gene copy number variation and RNAP copy number variation, the contribution of such “extrinsic” factors generally increases with mean gene expression, which are not directly captured in the pmf and thus will cause a deviation from the theoretical pmf.

Also, most of the distributions deviated from pmf in the first 1-2 bins. Why?

As the reviewer points out, a recurring pattern is observation of fewer cells with zero mRNA than the theoretical prediction, and more cells with one mRNA. We are not entirely sure why this is the case - one possibility is that the presence of partially transcribed or partially degraded mRNAs (which is not accounted for in our model) causes a greater proportion of cells to be identified as having one mRNA. Also, as mentioned above, there are certainly

other factors affecting the distributions (for instance, the detailed kinetics of promoter closed and open complex formation, or the influence of supercoiling on transcription kinetics) which make it unlikely that the theoretical distributions will fit perfectly at each mRNA copy number. We have added a brief discussion of these considerations to the main text.

In addition to including distribution-based fitting, it would be helpful to include more details about the robustness of the fitting for k_a and k_d . Possibly in the supplemental, can you explain if there are any parameter values that give you infinitely many solutions or no solutions?

Good question. In the following analysis, we take r and γ as givens i.e. independently measured constants. We identify two conditions necessary for a unique solution: (1) the mean expression μ must obey $0 < \mu < r/\gamma$ (2) The Fano factor F must obey between $1 < F < 1 + 1/(1 + K_D) * r/\gamma$, where we define $K_D = k_d/k_a$. If these conditions are met, there will be a unique solution, assuming that r and γ are fixed.

To see why, we find it helpful to think of this in two steps: (1) what does the mean expression tell us about the ratio k_d/k_a , and (2) what does the Fano factor (variability) tell us about the absolute magnitudes of k_d and k_a ? If we take r and γ as given, for any mean expression between 0 and r/γ , there is a value of k_d/k_a that satisfies equation 1 in the main text. This is simply another way of saying that the LacI is bound to the operator between 0 and 100% of the time. If we plot mean expression as a function of k_d and k_a , and imagine a line emanating from the origin to some point (k_d, k_a) we see that mean expression varies only as a function of the slope k_d/k_a of this line (Supplementary Fig. S5, upper left; this is a new figure). For convenience, we can define this slope as $K_D = k_d/k_a$ (analogous to the equilibrium dissociation constant).

From the mean expression, $K_D = k_d/k_a$ is fixed. We know that “correct” values are somewhere along the line with slope K_D , we just don’t know how far yet. Next we can plot the Fano factor as a function of k_a and k_d (Supplementary Fig. S5, upper right). Generally speaking, the Fano factor appears to decrease as we move away from the origin, although there is some fine-scale behavior near the origin (inset). If we use the fact that $k_a = k_d/K_D$, and that K_D is known from the mean expression, we can rewrite equation 2 in terms of k_d alone: $F = 1 + 1/(1 + K_D) * r/(\gamma + k_d/K_D + 1)$ and then plot how F depends on k_d for different values of K_D (lower right). We see that F is monotonically decreasing with increasing k_d . Interestingly, the value of F as $k_d \rightarrow 0$ does depend on K_D and is given by $1 + 1/(1 + K_D) * r/\gamma$; whereas $F \rightarrow 1$ as $k_d \rightarrow \infty$. This is consistent with the idea that slow kinetics are generally associated with more noise. As kinetics become much faster than γ , the distribution becomes more Poissonian and F approaches 1.

Thus, to recap, as long as μ is between 0 and r/γ , and F is between 1 and $1 + 1/(1 + K_D) * r/\gamma$, there will be a unique mathematical solution. We find it helpful to think of the solution geometrically whereby the (k_d, k_a) lies at the end of a line whose slope is dictated by the mean expression and whose length is dictated by the Fano factor; we hope that this interpretation is helpful to others. This has now been added as Supplementary Fig. 6.

Additionally, how sensitive is the fitting to the experimentally measured parameters? For example, if γ is off by 0.1 min⁻¹, how much would k_a and k_d vary.

Intuitively, γ sets the timescale for the entire system - when we speak of “fast” or “slow” binding kinetics, we implicitly mean “fast” or “slow” relative to the degradation rate γ .

Looking at equations 1 and 2 in the main text, we could have chosen to define $r' = r/\gamma$, $ka' = ka/\gamma$, $kd' = kd/\gamma$, that is to normalize/nondimensionalize all rates by dividing by γ , and then solve for ka' and kd' . In this case we would not have needed to measure γ directly since $r' = r/\gamma$ is simply read off from the $\Delta lacI$ experiments. These relative rates ka' and kd' would be independent of γ . So if e.g. $ka' = ka/\gamma$ is fixed by the data, a 10% larger value of γ would simply mean a 10% larger value of ka but there would be no effect on the relative ordering of ka and kd values.

Another way to see this is by looking at the analytic probability mass function (now equation 3 in the main text). The pmf depends only on the ratios of ka , kd , and r to γ (i.e. the nondimensionalized rates in terms of γ).

2. We are impressed by the level of kinetic detail obtained from the mRNA copy number distributions. If there are no technical limitations specific to this work, single molecule tracking could provide a more direct measurement of binding kinetics. Either the authors could compare their results to the kinetics measured in Elf et al 2007 (DOI: 10.1126/science.1141967) or could provide similar measurements for a few of the sequences tested in this work.

Elf et al 2007 measured association to the O1 operator with a LacI-venus fluorescent fusion protein, and determined the association rate ka to be 1 min^{-1} . In the current work, we found the equivalent rate to be 5.23 min^{-1} . However there are some differences that should be accounted for. First, Elf et al expressed LacI at 2-3 times lower concentration than wild-type (3-5 molecules per cell), so $ka = 1 \text{ min}^{-1}$ from Elf et al should be multiplied by a factor of 2-3 to be comparable. Second, the fluorescent fusion LacI cannot tetramerize whereas wt lacI does tetramerize. This means that the wt LacI can bind in twice as many configurations, and also can perform intersegmental transfer between non-contiguous DNA segments, so it seems plausible that this could lead to twice as large a ka value. So to compare ka from Elf et al 2007 to our measurement, we have $1 \text{ min}^{-1} \times \sim 2-3$ (expression) $\times 2$ (dimer vs tetramer) = $\sim 4-6 \text{ min}^{-1}$, which is consistent with our measured value of $ka = 5.23 \text{ min}^{-1}$.

For experiments like the one in Elf et al, low LacI-venus expression is a technical requirement to be able to detect single bound molecules against the fluorescent background, and there is no known way to create LacI fusions without abolishing dimerization, so we don't see any immediate possibility of improving on the results of Elf et al.

We have included a comparison to Elf et al 2007 in the main text.

3. The authors discussed that the in vivo measurements of ka and kd in *E. coli* are largely correlated with in vitro measurements in their previous publication. However, supplemental Fig. 7 mainly showed that they are proportional (repression ratio in vivo vs. relative occupancy in vitro. Fig. S7a), but the absolute values of ka and kd in vivo were not really correlated with in vitro measurement, especially for lacOsym (Fig. S7b, c). The discrepancy between the absolute values of ka/kd between in vivo and in vitro is expected because of the crowded cellular environment and the presence of large nonspecific binding sequences

nearby. Please discuss the implications of these differences—we find that the lacOsym being a synthetic sequence being less convincing.

In our response to reviewer #2, we have sought to explore in depth the possible reasons for the differences in *in vivo* vs *in vitro* kinetic rates, so we ask the reviewer to please refer to this response. We have also added relevant discussion to the main text.

4. It is significant to validate *in vivo* the major conclusion from the authors' previous work: the binding rate is dependent on the binding sequence, implying that the actual binding is determined by microscopic interactions after the initial bimolecular collisions. However, by this definition, shouldn't it be natural that k_a and k_d are anticorrelated since k_a includes the microscopic state interactions?

In hindsight we also think it's natural, and do not understand why it's been assumed that k_a is simply diffusion limited. We can speculate that it has been incorrectly assumed that evolution must have used the diffusion limited rate since search is so hard anyway. In fact, O. G. Berg, et.al, *Biochemistry* 20, 6929–6948 (1981) and O. G. Berg, C. Blomberg, *Biophysical Chemistry*, Volume 4, Issue 4, 1976, Pages 367-381 pointed out that diffusion limited association also implies that macroscopic dissociation is dependent on the microscopic association rate, but this observation has been left off from the biochemistry text books.

Additionally, the anticorrelation plot (Fig. 3) is mainly defined by lacOsym—if data points from lacOsym were taken out, the anticorrelation is much less significant. But lacOsym is also the one that appeared most different from *in vitro* measurement. Could the authors please clarify these points?

We acknowledge that the anticorrelation plot would be less significant without Osym. Since Osym is the strongest operator, it seems plausible that Osym and its mutants contain the most information about LacI kinetics in their mRNA distributions. We have added a comment to this effect in the main text.

Reviewer #5 (Remarks to the Author):

[Co-reviewer]

REVIEWERS' COMMENTS

Reviewer #1 (Remarks to the Author):

The authors have answered my concerns.

Reviewer #2 (Remarks to the Author):

The authors have addressed all the previously raised comments.

Reviewer #3 (Remarks to the Author):

Reviewer #3 (Remarks on code availability):

Didn't review the code

Reviewer #4 (Remarks to the Author):

We appreciate that the authors addressed most of our concerns in a much more thorough discussion section. Also, we find the addition of supplementary table 1 to be clear and concise and the addition of the analytic probability mass function to be helpful.

One minor concern is that whether the proportion of partially transcribed or degraded mRNAs would be indeed large enough to cause some of the very large 0 bins especially in the bottom row of supplementary figure 3. It is interesting to note that the 0 bin is overpopulated for the high expressing promoters, indicating that the pattern may not be due to stochastic burstiness. It is difficult to imagine such an association is specific for highly expressed promoters. Do the authors have any other explanations?

We appreciate the reviewer's thoughtful observation and we agree that something seems a bit off about the overpopulated zero bins. Our sense is that this is a technical/experimental issue affecting a few samples rather than a biological phenomenon. Our best guess is that some cells may not have been fully permeabilized, leading to inefficient FISH probe penetration and low FISH signal.

Reviewer #5 (Remarks to the Author):
